# Less Could Be More: Rethinking the Unexpected Deterioration of Variceal Bleeding After Endoscopic Occlusion of Gastroesophageal Varices

**DOI:** 10.3390/diagnostics15040461

**Published:** 2025-02-13

**Authors:** Ke Pang, Kun He, Yiyang Min, Zhiwei Wang, Dong Wu

**Affiliations:** 1State Key Laboratory of Complex Severe and Rare Diseases, Department of Gastroenterology, Peking Union Medical College Hospital, Chinese Academy of Medical Sciences and Peking Union Medical College, Beijing 100730, China; pangk19@student.pumc.edu.cn (K.P.); hk6290418@163.com (K.H.); minyy19@mails.tsinghua.edu.cn (Y.M.); 2Peking Union Medical College, Chinese Academy of Medical Sciences & Peking Union Medical College, Beijing 100730, China; 3State Key Laboratory of Complex Severe and Rare Diseases, Department of Radiology, Peking Union Medical College Hospital, Chinese Academy of Medical Sciences and Peking Union Medical College, Beijing 100730, China

**Keywords:** ectopic varices, cirrhosis, portal hypertension, gastrointestinal bleeding, endoscopy, multidisciplinary therapy

## Abstract

Ectopic varices account for 5% of variceal bleeding cases but carry high mortality due to their concealed nature and diagnostic challenges. A 46-year-old man with hepatitis C cirrhosis and prior gastroesophageal variceal bleeding presented with fatigue and dark red stools. Initial gastroscopy revealed moderate gastric–oesophageal varices without active bleeding, treated with preventive sclerotherapy and cyanoacrylate injection. Persistent bleeding and a worsening condition led to his transfer to our hospital. Clinical evaluation suggested lower gastrointestinal bleeding. Imaging and colonoscopy confirmed ascending colon ectopic varices with recent thrombotic bleeding, while a repeated gastroscopy showed evidence of prior therapeutic interventions for gastric–oesophageal varices, which were stable. A titanium clip was placed for temporary hemostasis, but further vascular embolization was halted due to extensive variceal involvement and risk of bowel necrosis. A multidisciplinary team recommended a transjugular intrahepatic portosystemic shunt, although the patient declined. This case underscores the importance of identifying the primary bleeding source to prevent exacerbation caused by unnecessary interventions. A stepwise diagnostic approach is put forward, highlighting that multidisciplinary care is crucial, with personalized, minimally invasive strategies to manage fragile vascular anatomy. Early detection and increased awareness of ectopic varices can facilitate timely and appropriate therapeutic interventions, ultimately improving patient care and outcomes.

**Figure 1 diagnostics-15-00461-f001:**
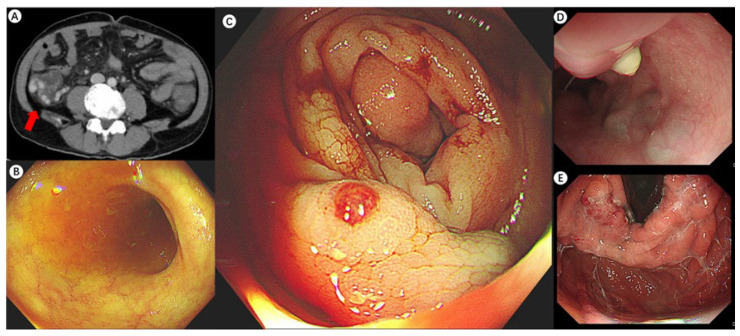
A 46-year-old man with hepatitis C-associated cirrhosis presented with severe fatigue and a one-month history of dark red bloody stools. A previous gastroscopy at a local hospital revealed moderate gastric–oesophageal varices without active bleeding. Endoscopic sclerotherapy and cyanoacrylate injection were performed as preventive measures. However, ongoing bleeding and a decline in the patient’s overall condition required his transfer to our hospital. Upon admission, his vital signs were at the lower limit of normal, and clinical examination showed anemia, mild jaundice, and active bowel sounds. Blood tests indicated low hemoglobin levels with normal urea nitrogen, and his Child–Pugh score was 11, consistent with advanced liver dysfunction. Based on the presence of dark red stools and normal urea nitrogen levels, the findings indicated lower gastrointestinal bleeding. Enhanced CT scan indicated tortuous thickening of blood vessels in the right abdominal cavity (the red arrow), raising suspicion of ectopic varices (**A**). An urgent colonoscopy provided additional clarity, revealing a clear view of the terminal ileum (**B**) and a submucosal protrusion with a thrombotic head in the ascending colon (**C**), confirming this as the site of the recent bleeding. A repeated gastroscopy showed evidence of prior therapeutic interventions for GOVs, which were stable (**D**,**E**). Notably, the patient had a prior episode of esophagogastric variceal bleeding one year earlier, which was successfully treated with ligation and cyanoacrylate injection. Evidence suggests that in the management of varices in portal hypertensive patients, precise identification of the bleeding target and minimizing unnecessary treatments may lead to more benefits for patients. Revealed risk factors of ectopic varices include portal hypertensive gastropathy, a history of abdominal and pelvic surgeries, and in this case, advanced Child–Pugh score and endoscopically eradicated esophageal varices. There is evidence suggesting that the impulsive occlusion of varices that is not responsible for active bleeding may increase the risk of hemorrhage from other sites. A study involving patients with PHT ectopic varices (*n* = 73) revealed that 57.9% of patients with ectopic varices had a history of treatment for esophageal varices, with an average interval of 2.8 ± 1.3 years in the rectum and 2.3 ± 2.1 years in the duodenum [1].

**Figure 2 diagnostics-15-00461-f002:**
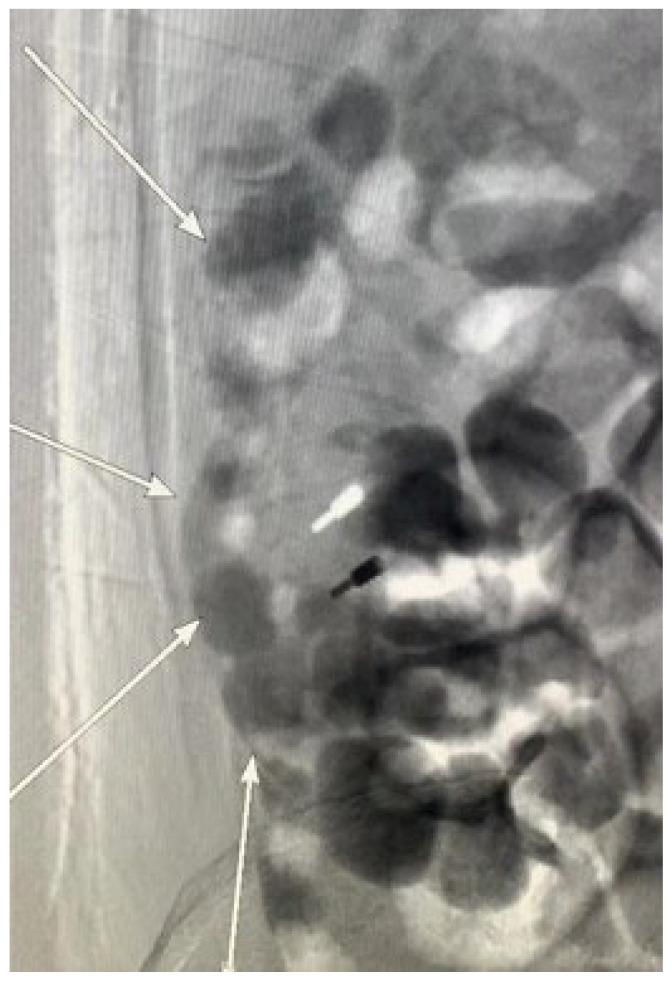
During the colonoscopy, a titanium clip was placed to achieve temporary hemostasis and provide time for vascular embolization under interventional radiology. Venography via the inferior vena cava performed to determine the extent of vascular involvement for embolization revealed multiple varicose veins in the ascending colon draining into the inferior vena cava (the white arrows). Due to the extensive variceal involvement and the high risk of congestion and necrosis following embolization, the vascular embolization procedure was discontinued. A multidisciplinary team evaluated the case; based on liver function, portal hypertension severity, and the condition of the varices, they recommended a transjugular intrahepatic portosystemic shunt (TIPS) procedure. The patient and their family declined this option, choosing instead to return to their local hospital for further management. Despite the interventions, the prognosis remained poor, with a significant risk of rebleeding due to decompensated liver cirrhosis, severe hepatic dysfunction, and extensive varices. The patient was advised to consult a specialized hepatic center for evaluation and potential liver transplantation. Six months later, during a follow-up phone call, the patient’s family reported that his condition was stable but declined to allow direct communication with the patient. Due to the lack of robust evidence-based treatment options for colonic variceal bleeding, the management of this patient was largely based on clinical experience and the medical resources available at our hospital. If varices are luminally accessible and stigmata of active or recent hemorrhage (i.e., white nipple and red signs) are present, first-line treatment should be endoscopic, either elastic band ligation (EBL) or endoscopic cyanoacrylate injection [2]. EBL has been successfully and safely used for managing duodenal and rectal variceal hemorrhage [3,4], although risk of rebleeding can arise depending on the location, size, and accessibility of the varices, with some reports of a 40% rebleeding rate in patients with rectal varices [5]. Endoscopic cyanoacrylate injection is also effective in achieving hemostasis and preventing rebleeding in duodenal varices, with good long-term outcomes. However, studies regarding its use in colorectal varices is still limited [6]. Endoscopic ultrasound-guided (EUS) therapies, including EUS-guided cyanoacrylate injection [7] and coil embolization [8] are emerging as important tools for variceal management. Coil embolization, in particular, can provide a scaffold for injecting smaller volumes of cyanoacrylate glue in a combined approach. These techniques offer significant advantages, including real-time Doppler assessment of variceal flow and the ability to identify varices that may not be visible via mucosal inspection. However, the current application of EUS-guided variceal interventions is largely limited to duodenal [9] or rectal varices [10,11], with only limited reports on their use for proximal colonic varices.

**Figure 3 diagnostics-15-00461-f003:**
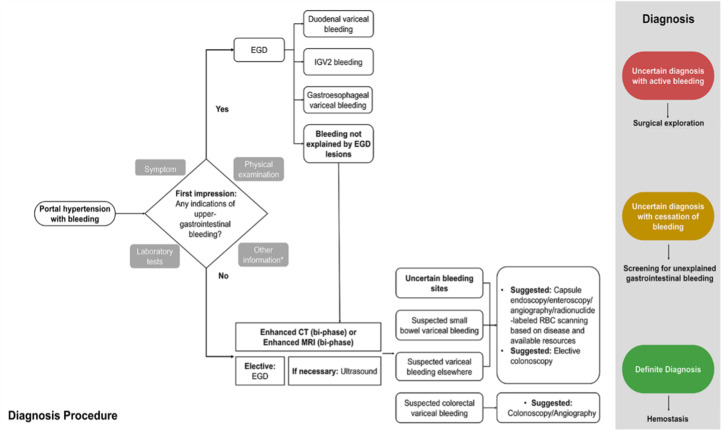
Diagnosis algorithm for ectopic variceal bleeding in portal hypertension (* Other information: Any relevant details, including the patient’s medical history). Currently, there is no consensus in the academic community on the diagnosis of ectopic varices. Therefore, the authors propose a stepwise diagnostic approach. Symptom-based assessment of bleeding origin determines the sequence and approach of endoscopic screening [12]. When there are no signs of active gastroesophageal variceal bleeding to account for circulatory failure, worsening anemia, dark red bloody stools, increased bowel sounds, and elevated blood urea nitrogen, suspicion of lower gastrointestinal bleeding should be immediately considered. Performing timely enhanced CT or enhanced MRI with targeted colonoscopy during the initial visit could have facilitated prompt intervention and stopped the bleeding. Given the rarity and concealed nature of ectopic varices, identifying the precise focus of hemostasis, however, can be challenging [13]. The diagnosis of ruptured bleeding from gastroesophageal varices is typically established through the observation of specific conditions during endoscopy: active bleeding within the varices, varices overlying a platelet–fibrin plug (known as white nipple sign), or varices overlaid with blood clots without other bleeding sources [14]. In contrast, duodenal variceal bleeding, for instance, is often submucosal and located on the convex side outside the lumen, making it difficult to detect during endoscopy [15]. Invasive methods, like enteroscopy, endoscopic ultrasound, and angiography, as well as noninvasive methods, such as video capsule endoscopy, contrast-enhanced imaging techniques, Doppler ultrasound, and radionuclide scanning, may be necessary for a definitive diagnosis. Surgical intervention may be considered if other diagnostic approaches are inconclusive. Due to the limited literature, determining site-specific treatments is challenging. Each site presents unique factors, including etiology, vascular anatomy, bleeding risk severity, hepatic reserve, and other variables, requiring personalized management. Therefore, a multidisciplinary approach involving endoscopists, interventional radiologists, and surgeons is strongly recommended. This case highlights the need to consider ectopic varices as a potential cause of atypical bleeding in patients with portal hypertension. Differentiating upper and lower gastrointestinal bleeding is one of the ABCs in diagnosing gastrointestinal diseases. In clinical practice, it is crucial not to overlook the interpretation of physical examination findings and laboratory results but solely rely on empirical knowledge, such as the common occurrence of variceal bleeding in the gastroesophageal region. Neglecting these aspects, as seen in this case, can result in delayed hemostasis and unnecessary, or even harmful invasive treatments. Early detection and an increased awareness of ectopic varices can facilitate timely and appropriate therapeutic interventions, ultimately improving patient care and outcomes. For the treatment of ectopic varices in patients with portal hypertension, less could indeed be more.

## Data Availability

No new data were created or analyzed in this study.

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
