# Peer review of "Less Could Be More: Rethinking the Unexpected Deterioration of Variceal Bleeding After Endoscopic Occlusion of Gastroesophageal Varices"

_diagnostics, 2025, doi:10.3390/diagnostics15040461_

Round 1

Reviewer 1 Report

Comments and Suggestions for Authors

The author's provides a very interesting case with interesting images which were done in a patient with ectopic variceal bleeding. It is also very well done that case-report, with important algorithms to detect the bleeding of ectopic varices. The article must be accepted in the current form.

Author Response

Comment: The author's provides a very interesting case with interesting images which were done in a patient with ectopic variceal bleeding. It is also very well done that case-report, with important algorithms to detect the bleeding of ectopic varices. The article must be accepted in the current form.

Response: We sincerely appreciate your kind and positive feedback. We are grateful for your recognition of the case and the algorithms, and we are pleased that the manuscript met your expectations.

Reviewer 2 Report

Comments and Suggestions for Authors

The paper is very interesting due to the rarity of the case. 

Although, the presentation is quite strange and the case is described in the figure legends instead of a proper introduction and case discussion 

I suggest you to change it.

Reviewer 3 Report

Comments and Suggestions for Authors

This study presents a case of ectopic variceal bleeding in a patient with hepatitis C cirrhosis, highlighting the diagnostic challenges and management dilemmas associated with this condition. The authors emphasize the potential risks of overtreatment, particularly the exacerbation of bleeding when non-culprit varices are occluded. They advocate for a stepwise diagnostic approach and multidisciplinary management to optimize patient outcomes.

The manuscript has several language inconsistencies that affect readability. One particularly problematic sentence in the abstract states: "This case highlights the importance of identifying the victim vessels to avoid bleeding exacerbation led by overtreatment." The term "victim vessels" is awkward and unclear. A better alternative might be: "This case underscores the importance of identifying the primary bleeding source to prevent exacerbation caused by unnecessary interventions."

Additionally, some sentences are overly complex or unclear. For example, the phrase: "It’s not without evidence that impulsive occlusion of non-victim varices could increase the risk of bleeding from other varices." is convoluted. A clearer version would be: "There is evidence suggesting that the impulsive occlusion of varices not responsible for active bleeding may increase the risk of hemorrhage from other sites."

  • The manuscript lacks sufficient elaboration on the technique of elastic band ligation (EBL) in the colon or rectum. While this technique is widely used for esophageal varices, its application in lower gastrointestinal varices is less well-documented. The authors should discuss technical considerations, success rates, and complications specific to EBL in the colon.

  • The discussion of embolization risks is insightful, but more detail on alternative hemostatic approaches, such as endoscopic ultrasound-guided therapy or sclerotherapy for colonic varices, would enhance the clinical relevance.

  • There is some inconsistency in terminology. The manuscript refers to both "portal gastropathy" and "portal hypertension gastropathy"—standardizing to one term would improve clarity.

Comments on the Quality of English Language

The manuscript has several language inconsistencies that affect readability

Author Response

Thank you very much for you valuable advice. Please see the attachment.

Round 2

Reviewer 2 Report

Comments and Suggestions for Authors

I am happy after tour revisions

Best regards